# Solution-Processed OLED Based on a Mixed-Ligand Europium Complex

**DOI:** 10.3390/ma16030959

**Published:** 2023-01-19

**Authors:** Makarii I. Kozlov, Kirill M. Kuznetsov, Alexander S. Goloveshkin, Andrei Burlakin, Maria Sandzhieva, Sergey V. Makarov, Elena Ilina, Valentina V. Utochnikova

**Affiliations:** 1Department of Chemistry, M.V. Lomonosov Moscow State University, 1/3 Leninskie Gory, 119991 Moscow, Russia; 2Department of Material Sciences, M.V. Lomonosov Moscow State University, 1/3 Leninskie Gory, 119991 Moscow, Russia; 3A. N. Nesmeyanov Institute of Organoelement Compounds, Vavilova St. 28, 119334 Moscow, Russia; 4School of Physics and Engineering, ITMO University, Lomonosova 9, 197101 St. Petersburg, Russia; 5Qingdao Innovation and Development Center, Harbin Engineering University, Qingdao 266000, China; 6Institute of Chemistry and Chemical-Pharmaceutical Technologies, Altai State University, Prospekt Lenina 61, 656049 Barnaul, Russia

**Keywords:** OLED, electroluminescence, europium, mixed-ligand complex, lifetime

## Abstract

An approach to increase the efficiency of europium-based OLEDs was proposed through the formation of a mixed-ligand complex. The design of a series of europium complexes, together with an optimization of the solution deposition, including the host selection, as well as the variation of the solvent and deposition parameters, resulted in a noticeable increase in OLED luminance. As a result, the maximum luminance of the Eu-based OLED reached up to 700 cd/m^2^, which is one of the highest values for an Eu-based solution-processed OLED. Finally, its stability was investigated.

## 1. Introduction

Despite the rapid development of organic light-emitting diode (OLED) technology, several important issues remain to be solved. Among them, narrowing the luminescence bands of the OLEDs is particularly important [1] due to the swift acceleration of wearable electronics development [2,3], including in tissue oximetry [4], where narrow-band luminescence simplifies the detection and fits into the transparency window of biological tissues [5]. The ultimate solution to this problem is the use of lanthanide coordination compounds (Ln CCs) with extremely narrow luminescence bands (<10 nm) [6]. In particular, europium complexes with their high quantum yields (PLQYs) and red emission (~615 nm) are of interest. However, the peculiarities of Ln-based OLEDs [7], particularly their long exciton lifetime (τ) [8], still result in low luminance and efficiencies of OLEDs based on Eu CCs [9]. This makes it an urgent task to search for new materials, as well as to optimize the OLED deposition parameters, to which Eu CCs with their long lifetimes are particularly sensitive.

Most commercially available OLEDs are produced using vacuum thermal evaporation (VTE) due to the higher luminance and efficiency of OLEDs produced this way, resulting from the higher purity of the deposited films [10]. At the same time, VTE has some disadvantages, including the impossibility of using non-volatile compounds, high complexity, cost of the technological process, and the complexity of optimization of the material co-evaporation [11]. Thus, the development of solution-processed methods is itself an important task. 

However, solution deposition is always associated with fast degradation [10,12,13], which is particularly harmful to Ln-based OLEDs, whose long excited-state lifetimes already facilitate degradation. As a result, the highest luminance obtained for solution-processed lanthanide-based OLEDs is very low, while their degradation has not been studied at all.

At the same time, for some materials, it has already been possible to obtain spin-coated (SC) OLEDs, which are only slightly inferior to similar VTE devices [13], due to the SC film deposition process, i.e., the optimal solvent, rotation rate, and annealing conditions. Mao et al. reported similar OLED performance for VTE-deposited and SC films [14], while Feng et al. reported that SC films of TPD (tri-methylphenyl diamine) were superior to the VTE-deposited ones, being much smoother and denser [15]. For Ln-based OLEDs, this was rarely studied, but we have recently demonstrated [16] that solution-deposited films of Eu CCs are denser than VTE-deposited films, which resulted in a decrease in the turn-on voltage.

In this paper, SC deposition of the emission layers based on europium mixed-ligand complexes was studied and optimized. Their photophysical properties, depending on the host material and deposition conditions, were studied, aiming at obtaining the highest PLQY/τ ratio, and the optimized films were tested in OLEDs. As the object of study, mixed-ligand europium complexes Eu(dik)_3_DPPZ with dipyrido [3,2-a:2′c,3′c-c]phenazine (DPPZ) β-diketones (dik = tta (thenoyltrifluoroacetone), dbm (dibenzoylmethane), and btfa (benzoyltrifluoroacetone)) were used, since DPPZ-containing β-diketonates demonstrated the record luminance of electroluminescence (EL) [17].

## 2. Materials and Methods

All solvents and chemicals were purchased from commercial sources.

Powder X-ray diffraction data (PXRD) were collected using Bruker D8 Advance (Karlsruhe, Germany) [λ(Cu-Kα) = 1.5418 Å; Ni filter] with a step size of 0.020°.

A suitable single crystal of [Eu(btfa)_3_DPPZ] (C_48_H_28_EuF_9_N_4_O_6_) was selected and mounted on a Bruker Quest diffractometer. The crystal was kept at 100 K during data collection. Using Olex2 [18], the structure was solved with the XS [19] structure solution program using Direct Methods and refined with the XL [19] refinement package using Least Squares minimization.

Crystal Data for [Eu(btfa)_3_DPPZ] (C_48_H_28_EuF_9_N_4_O_6_, M = 1079.70 g/mol): tetragonal, space group *P*4/*n* (no. 85), a = 27.5502(3) Å, c = 11.6190(3) Å, V = 8819.0(3) Å^3^, Z = 8, T = 100 K, μ(MoKα) = 1.515 mm^−1^, D_calc_ = 1.626 g/cm^3^, 52,490 reflections measured (3.804° ≤ 2Θ ≤ 56.584°), 10,940 unique (R_int_ = 0.0624, R_sigma_ = 0.0531), which were used in all calculations. The final R1 was 0.0563 (I > 2σ(I)) and wR2 was 0.1235 (all data).

CCDC 2223863 contains the supplementary crystallographic data for this paper. These data can be obtained free of charge from The Cambridge Crystallographic Data Centre via http://www.ccdc.cam.ac.uk (deposition date 2 December 2022).

Thermal analysis was carried out on an STA 409PC Luxx thermoanalyzer (NETZSCH, Selb, Germany) in the temperature range of 20–1000 °C in air, at a heating rate of 10 (°)/min. The evolved gases were simultaneously monitored during the TA experiment using a coupled QMS 403C Aeolos quadrupole mass spectrometer (NETZSCH, Selb, Germany). The mass spectra were registered for the species with the following *m*/*z* values: 18 (corresponding to H_2_O), 44 (corresponding to CO_2_), 46 (corresponding to C_2_H_5_OH), and 127 (corresponding to I).

The IR spectra were recorded on a Nicolet iS50 FTIR Spectrometer as a powder at ATR (Thermo Scientific, Waltham, MA, USA).

Photoluminescence spectra were recorded using a Fluoromax Plus (HORIBA, Piscataway, NJ, USA) spectrometer at room temperature; excitation was performed through a ligand, and the absolute method in the integration sphere was used.

### 2.1. OLED Manufacture

Prepatterned indium tin oxide coating with 15 Ohm/sq on the glass substrates (Kaivo LTD) were used as anodes. The substrates were cleaned by three-step ultrasonication in deionized water, acetone, and isopropanol for 15 min each followed by drying with airflow. Then a 20 min UV treatment was performed to remove residual organic impurities.

Hole injection layer PEDOT-PSS (poly(3,4-ethylenedioxythiophene):poly(styrenesulfonate) (Ossila Al-4083) was spin-coated on cleaned ITO glass substrates at 3000 rpm for 1 min with the following annealing process at 140 °C for 10 min in the air. Then a 20 nm thick hole-transporting poly-TPD (Ossila) solution was spin-coated from 5 mg/mL solution in chlorobenzene at 1500 rpm for 1 min and dried at 120 °C for 10 min in the nitrogen-filled glovebox. Afterward, a 30 nm thick emission layer was spin-coated from THF (Eu(dik)_3_DPPZ:CBP 1:3, total c = 5 g·L^−1^) at 1500 rpm for 1 min with further annealing at 80 °C for 10 min.

Finally, the substrates were transferred into a MB-ProVap 5G vacuum deposition system. The ∼20 nm thick electron-transporting/hole-blocking layer TPBi (Lumtec) was thermally evaporated followed by a ∼1 nm thick LiF layer and 100 nm thick aluminum layer as the cathode in a sequence through a shadow mask at 10^−6^ mbar to form 21 mm^2^ pixels. The thicknesses of all evaporated layers was controlled by a quartz micro-balance resonator pregraduated by profilometry.

Measurements of the OLED characteristics were performed in the N_2_ glovebox without encapsulation. The electroluminescence spectra were obtained with an Instrument Systems CAS 120 Array spectrometer sensitive within 200–1100 nm. Current-voltage characteristics were measured by using Keithley 2400 source-meter measurement unit. The turn-on voltage was defined as the voltage at which 1 cd/m^2^ EL intensity was achieved.

### 2.2. Synthesis

Synthesis of Eu(dbm)_3_DPPZ. A solution of 1 mmol of EuCl_3_·6H_2_O in 20 mL of ethanol was added to a mixture of Hdbm (3 mmol), Et_3_N (3 mmol) in 20 mL of ethanol, then a solution of DPPZ (1 mmol) in 30 mL of ethanol was added, and the precipitation was observed. The reaction mixture was stirred for 2 h, then the precipitate was filtered off, washed with cold ethanol, and dried in air.

Synthesis of Eu(tta)_3_DPPZ. A solution of 1 mmol of EuCl_3_·6H_2_O in 20 mL of ethanol was added to a mixture of Htta (3 mmol), Et_3_N (3 mmol) in 20 mL of ethanol, then a solution of DPPZ (1 mmol) in 30 mL of ethanol was added, and the precipitation was observed. The reaction mixture was stirred for 2 h, then the precipitate was filtered off, washed with cold ethanol, and dried in air.

Synthesis of Eu(btfa)_3_DPPZ. A solution of 1 mmol of EuCl_3_·6H_2_O in 20 mL of ethanol was added to a mixture of Hbtfa (3 mmol), Et_3_N (3 mmol) in 20 mL of ethanol, then a solution of DPPZ (1 mmol) in 30 mL of ethanol was added, and the precipitation was observed. The reaction mixture was stirred for 2 h, then the precipitate was filtered off, washed with cold ethanol, and dried in air.

A single crystal of Eu(btfa)_3_DPPZ was obtained by the slow evaporation of the Eu(btfa)_3_DPPZ solution in ethanol at room temperature.

## 3. Results and Discussion

### 3.1. Synthesis and Characterization

Mixed-ligand complexes Eu(dik)_3_DPPZ were synthesized as in [17]. A single crystal of [Eu(btfa)_3_DPPZ] was obtained by slow evaporation of the ethanol solution of Eu(btfa)_3_DPPZ (Figure 1b). The complex is monomeric, and the Eu^3+^ central ion is surrounded by three btfa^−^ anionic ligands and one DPPZ neutral ligand, which results in CN = 8.

The PXRD pattern of Eu(btfa)_3_DPPZ coincided with the one calculated from the [Eu(btfa)_3_DPPZ] structure (Figure 2b); the PXRD patterns of the other Eu(dik)_3_DPPZ (dik = tta, dbm) coincided with those calculated from the structures of [Eu(tta)_3_DPPZ] and [Dy(dbm)_3_DPPZ] (CCDC numbers 2072867 [16] and 1062453, Appendix A). In order to additionally confirm the composition of complexes, ^1^H NMR spectroscopy, TGA, and IR spectroscopy were performed (Figure 2a, Appendix A).

### 3.2. Photoluminescent Properties

We studied photoluminescent (PL) properties of the complexes’ thin films, as well as of the composite films Eu(dik)_3_DPPZ:host (Table 1), where the following hosts were selected based on the literature data: 6TCTA:3OXD-7, 6CBP:3OXD-7, 6TCTA:3CBP, and CBP itself [20,21,22]. Here TCTA is 4,4′,4″-tris(carbazol-9-yl)triphenylamine, OXD-7 is 1,3-bis[2-(4-tert-butylphenyl)-1,3,4-oxadiazo-5-yl]benzene, and CBP is 4,4′-Bis(N-carbazolyl)-1,1′-biphenyl.

Both doped and undoped Eu(dik)_3_DPPZ films exhibit photoluminescence typical of europium ions; the organic photoluminescence is absent (Figure 3 and Appendix A). The excitation spectra demonstrate broad excitation bands through the dik^−^ (250–300 nm) and the DPPZ (300–400 nm) and are almost unaffected by the host [23].

The PLQY and observed lifetime (τ_obs_) values strongly depend on the host. This demonstrates the importance of the measurements of the photophysical data within the composite film even if the properties of the pure compound are known. Such dependence may be associated with the participation of the host in the sensitization process. Indeed, as DPPZ demonstrates low sensitization efficiency and even quenching of the Eu photoluminescence, the presence of the host, able to sensitize its photoluminescence, may affect the photoluminescence efficiency.

At the same time, as both the lifetimes and the PLQY increase, we calculated the PLQY/τ_obs_ parameter to compare the obtained films. Indeed, the increase of the PLQY linearly increases the electroluminescence intensity, while the increase of the τ_obs_ results in its linear decrease. Thus, the larger PLQY/τ_obs_ is, the better should be the performance. Interestingly, the PLQY/τ_obs_ value did not change much after the doping, though it increases in some hosts, and decreases in others. This data demonstrates that according to the photophysical characteristics, CBP can be considered the best host material among the selected ones.

For CBP, as well as for CBP:TCTA, we also studied the dependence of the composite films’ photophysical properties on the solvent, from which it was deposited (Table 2). We selected dichloromethane (DCM), tetrahydrofuran (THF), and toluene (Tol) as the most suitable solvents for solution deposition according to the literature data. The photophysical properties’ dependence on the dopant concentration was also studied.

It revealed that the PLQY values, as well as the τ_obs_ and PLQY/τ_obs_ values, do indeed strongly depend on the solvent. Thus, the PLQY and PLQY/τ_obs_ values are lower in the high-boiling toluene than in DCM and THF. This is likely due to the quenching by the residual solvent, which remains within the film even after the thermal treatment.

The comparison of the obtained data demonstrates that the best performance is demonstrated by the Eu(tta)_3_DPPZ:CBP = 1:3 film, deposited from THF.

### 3.3. OLED Fabrication

Based on the obtained data, OLEDs S1–S3 with the heterostructure ITO/PEDOT:PSS/poly-TPD/**EML**/TPBi/LiF/Al (TPBi = 2,2′,2″-(1,3,5-benzinetriyl)-tris(1-phenyl-1-H-benzimidazole)) were obtained (Figure 4), where spin-coated Eu(dik)_3_DPPZ:CBP 1:3 films served as the EML (dik = tta for S1, btfa for S2, dbm for S3).

All the diodes demonstrated the pure EL of the europium ion (Appendix A) with CIE coordinates (x = 0.66, y = 0.32), which coincided exactly with their photoluminescence spectra (Figure 3); their luminance reached up to 700 cd/m^2^ (Figure 5). This is one of the highest values obtained so far for solution-deposited Eu-based OLEDs, and it is a result of the purposeful selection of not only the emitter but also the deposition conditions.

OLED degradation and stability is an important issue, which has never been studied for lanthanide-based OLEDs. In the present work, the stability was studied for the OLED S1 at 7 V and 8.6 V (starting efficiency of ca. 10 cd/m^2^ and 100 cd/m^2^, respectively).

At both voltages, an interesting phenomenon was observed (Figure 6): upon measurement, first, an increase in intensity was observed, which reached 33% at 7 V; after 25 s, the intensity increased from 14 cd/m^2^ to 18.66 cd/m^2^. This is quite unusual and may be connected with the further solvent elimination from the layer or heating up of the OLED device. Thus, the obtained maximum intensity, t_50_ of 10,643 s = 177.3 min~3 h, was measured, while from the initial value of 14 cd/m^2^, t_50_ has not even been reached after 12,000 s.

At 8.6 V, the intensity increase is not that pronounced. From the initial intensity, t_50_ of 1971 s~33 min was reached. These values are quite remarkable for the lanthanide-based solution-processed OLEDs.

## 4. Conclusions

We demonstrated the strong dependence of the photophysical properties of composite films containing europium complexes on the composition, such as host and doping concentration, as well as deposition parameters. The optimization of these parameters allowed us to obtain the films with the highest PLQY/τ ratio, i.e., films doped into CBP, deposited from the THF. These films were tested in OLEDs, and a luminance of up to 700 cd/m^2^ was obtained thanks to the purposeful selection of the deposition conditions, with t_50_~33 min at 100 cd/m^2^. This is the first study of lanthanide-based OLED stability.

## Figures and Tables

**Figure 1 materials-16-00959-f001:**
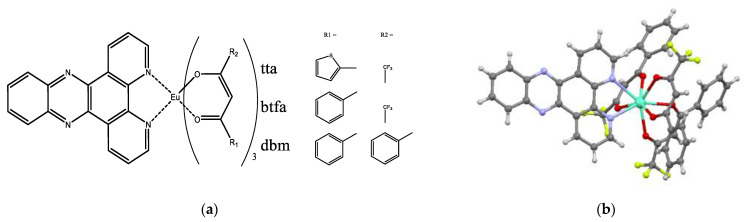
(**a**) Complexes Eu(dik)_3_DPPZ used in the present study. (**b**) The structure of [Eu(btfa)_3_DPPZ] in crystal.

**Figure 2 materials-16-00959-f002:**
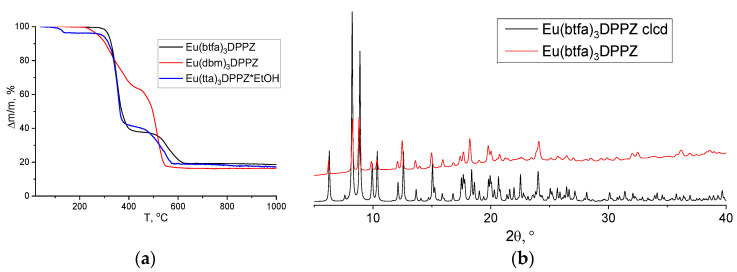
(**a**) TGA data of Eu(btfa)_3_DPPZ, Eu(tta)_3_DPPZ·EtOH, and Eu(dbm)_3_DPPZ powders and (**b**) PXRD patterns for Eu(btfa)_3_DPPZ: experimental (red curve) and calculated from the crystal structure (black curve).

**Figure 3 materials-16-00959-f003:**
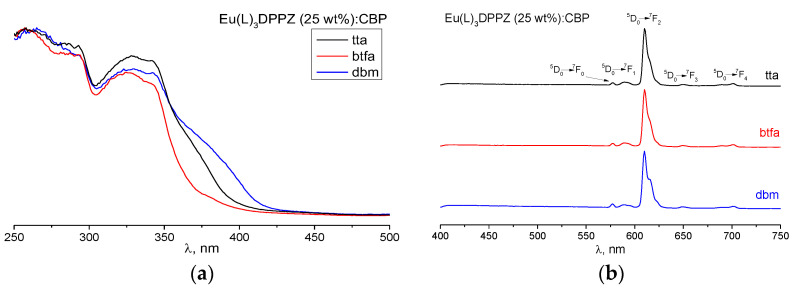
(**a**) Excitation and (**b**) photoluminescence spectra of Eu(dik)_3_DPPZ thin films doped into CBP: Eu(dik)_3_DPPZ, dik = dbm (blue), btfa (red), and tta (black).

**Figure 4 materials-16-00959-f004:**
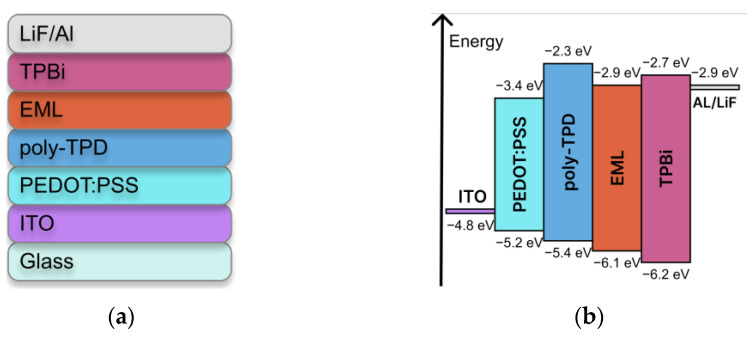
(**a**) Heterostructure and (**b**) band structure of OLEDs.

**Figure 5 materials-16-00959-f005:**
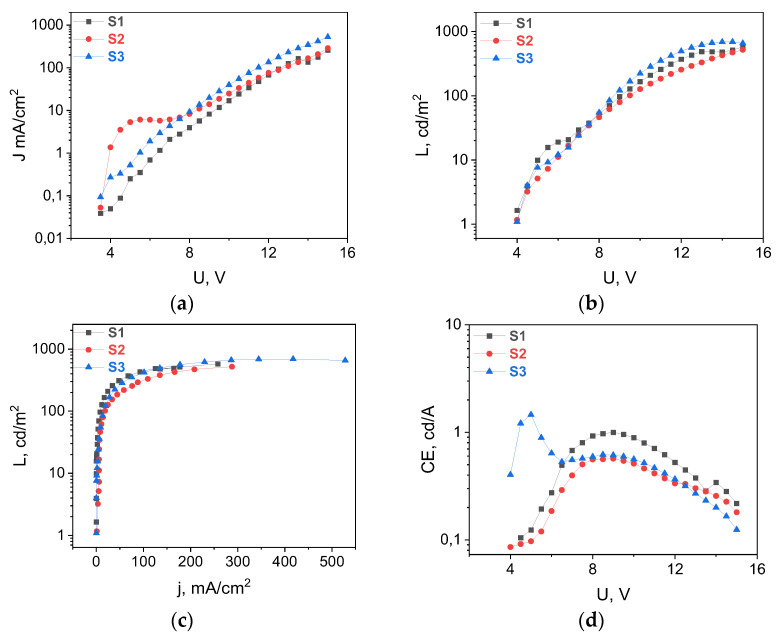
(**a**) J-V, (**b**) L-V, (**c**) L-J, and (**d**) CE-V curves of Eu(dik)_3_DPPZ:CBP 1:3 thin films: dbm (blue), btfa (red), and tta (black).

**Figure 6 materials-16-00959-f006:**
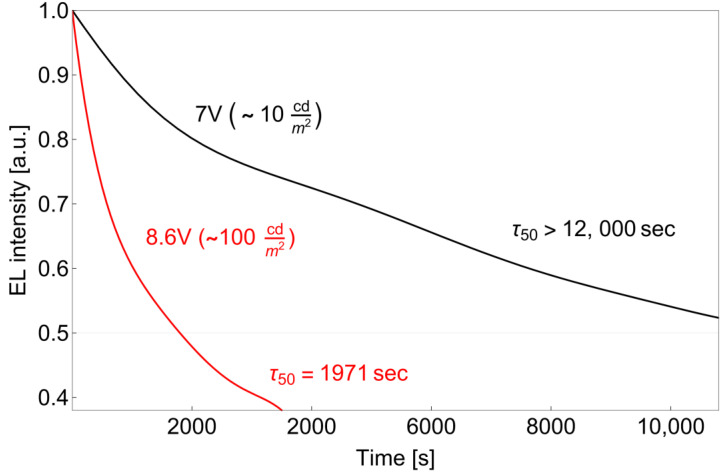
Lifetime of solution-processed OLED based on Eu(dbm)_3_DPPZ under the constant voltage of 7 V (~10 cd/m^2^) and 8.6 V (~100 cd/m^2^).

**Table 1 materials-16-00959-t001:** Photoluminescent properties of the Eu(dik)_3_DPPZ:host films (λ_ex_ = 330 nm).

dik =	Host in Eu(dik)_3_DPPZ:host	PLQY, %	τ_obs_, ms	PLQY/τ_obs_
tta	-	0.4	0.07	6
	6TCTA:3OXD-7	2.3	0.25	9
	6CBP:3OXD-7	6.4	0.37	17
	6TCTA:3CBP	10.7	0.34	32
	**3CBP**	**11**	**0.24**	**46**
btfa	-	1.3	0.10	13
	6TCTA:3OXD-7	2.2	0.24	9
	6CBP:3OXD-7	5.4	0.39	14
	6TCTA:3CBP	4.9	0.31	16
	**3CBP**	**4.6**	**0.27**	**17**
dbm	-	0.6	0.06	10
	6TCTA:3OXD-7	2.3	0.21	11
	6CBP:3OXD-7	5.1	0.31	16
	6TCTA:3CBP	4.1	0.26	16
	**3CBP**	**3**	**0.17**	**18**

**Table 2 materials-16-00959-t002:** The solvent dependence of the photophysical properties of the Eu(tta)_3_DPPZ-doped films.

Solvent	Eu(tta)_3_DPPZ:CBP:TCTA	PLQY, %	τ_obs_, ms	PLQY/τ_obs_	Eu(tta)_3_DPPZ:CBP	PLQY, %	τ_obs_, ms	PLQY/τ_obs_
DCM	1:(7/3):1	4.0	0.20	20	1:1	2.5	0.15	17
	1:3.5:1.5	7.0	0.29	24	1:2	2.7	0.18	15
	1:7:3	9.7	0.34	29	1:3	3.0	0.20	15
THF	1:(7/3):1	5.0	0.24	21	1:1	4.3	0.18	24
	1:3.5:1.5	9.8	0.33	30	1:2	8.2	0.21	39
	1:7:3	10.7	0.34	32	1:3	11.0	0.24	46
Tol	1:(7/3):1	2.2	0.14	16	1:1	1.6	0.10	16
	1:3.5:1.5	3.6	0.21	17	1:2	3.9	0.16	24
	1:7:3	4.8	0.26	19	1:3	5.3	0.19	28

## Data Availability

Data available on request.

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
