# Peer review of "Solution-Processed OLED Based on a Mixed-Ligand Europium Complex"

_materials, 2023, doi:10.3390/ma16030959_

Round 1

Reviewer 1 Report

Report

In this work, mixed-ligand europium complexes Eu(dik)3DPPZ with dipyrido[3,2-a:2’c,3’c- 63 c]phenazine (DPPZ) and b-diketonates (dik = tta (thenoyltrifluoroacetone), dbm (diben- 64 zoylmethane), and btfa (benzoyltrifluoroacetone)) were synthesized to use them in the study of the emission layers depending on the host material, on the deposition (SC and VTE) and the solvent to increase the OLED luminance and efficiency.

Therefore, in my opinion, the manuscript can be published after some major revision as follow:

I find Figure S3 very informative. Nevertheless, the pattern of the experimental XRPD of Eu(dbm)3DPPZ and the theoretical XRPD diffractogram with Dy are different, which means that we do not have the same complex. At least, till 30 º in 2θ we must have very similar diffractograms.  For example, if we have a look around 17 º in 2θ, we can observe that data do not match and it is not only a slight displacement due to the difference of the temperature, logical difference, as we can observe for the first peak. In conclusion, we do not know which complex is really used.

Regarding the indexed XRPD pattern of Eu(tta)3DPPZ·EtOH, more information is needed. Please, do.

Please, in Figure S3 a homogeneous legend is required: 2θ or 2 theta with the corresponding unit (º).

Please, btfa, DPPZ, TCTA… must be specified the first time there are used.

In the synthesis of Eu(btfa)3DPPZ, the obtention of suitable single crystals for XR diffraction is not described at all, only the powder (precipitate).

In line 149, references for both structures must be given.

A space between values and units must be respected, please revise the manuscript.

Concerning the crystal data, the letters for the space group must be written in italic: P4/n

Author Response

Dear Reviewer,

we thank you for your attentive reading and helpful comments on this work. Below is a list of changes that we made based on your suggestions.

  1. 1. I find Figure S3 very informative. Nevertheless, the pattern of the experimental XRPD of Eu(dbm)3DPPZ and the theoretical XRPD diffractogram with Dy are different, which means that we do not have the same complex. At least, till 30 º in 2θ we must have very similar diffractograms.  For example, if we have a look around 17 º in 2θ, we can observe that data do not match and it is not only a slight displacement due to the difference of the temperature, logical difference, as we can observe for the first peak. In conclusion, we do not know which complex is really used.

Thank you for noticing! Indeed, they did not coincide, but the pattern of Eu(dbm)3DPPZ was successfully Pawley fitted with monoclinic cell based on Dy(dbm)3DPPZ structure (CCDC 1062453). The information is added to the ESI.

Fig.S3. a) Indexed PXRD pattern of Eu(tta)3DPPZ·EtOH, b) Pawley fit (red line) of Eu(dbm)3DPPZ (blue line) and their difference (grey curve). c,d) Molecules in the structures of c) Eu(tta)3DPPZ and
d) Eu(btfa)3DPPZ.

Obtained pattern of Eu(dbm)3DPPZ was successfully Pawley fitted with monoclinic cell based on Dy(dbm)3DPPZ structure (CCDC 1062453). The cell parameters after refinement are following:

   R-Bragg                                   0.175

   Spacegroup                                P21/c

   Cell Volume (Å3)                         10257(9)

   Lattice parameters

      a (Å)                                  12.289(7)

      b (Å)                                  20.829(10)

      c (Å)                                  40.716(19)

      beta  (°)                              100.20(4)

  1. 2. Regarding the indexed XRPD pattern of Eu(tta)3DPPZ·EtOH, more information is needed. Please, do.

Thank you, added to the ESI:

Powder pattern of Eu(tta)3DPPZ·EtOH was successfully Rietveld refined using the [Eu(tta)3DPPZ]·MeCN structure (CCDC 2072867). Only the cell parameters were optimized, the atomic coordinates were taken from the structure of the single crystal and were not refined.

  1. 3. Please, in Figure S3 a homogeneous legend is required: 2θ or 2 theta with the corresponding unit (º).

Done

  1. 4. Please, btfa, DPPZ, TCTA… must be specified the first time there are used.

btfa is specified in line 65, DPPZ – in line 63.

Host name specifications were added:

Here TCTA is 4,4',4"-tris(carbazol-9-yl)triphenylamine, OXD-7 is 1,3-bis[2-(4-tert-butylphenyl)-1,3,4-oxadiazo-5-yl]benzene, and CBP is 4,4′-Bis(N-carbazolyl)-1,1′-biphenyl

  1. 5. In the synthesis of Eu(btfa)3DPPZ, the obtention of suitable single crystals for XR diffraction is not described at all, only the powder (precipitate).

Added:

Single crystal of Eu(btfa)3DPPZ was obtained by the slow evaporation of the Eu(btfa)3DPPZ solution in ethanol at room temperature.

  1. 6. In line 149, references for both structures must be given.

Done

  1. 7. A space between values and units must be respected, please revise the manuscript.

Done

  1. 8. Concerning the crystal data, the letters for the space group must be written in italic: P4/n

Done

Reviewer 2 Report

Authors report the europium complex compounds, their properties and their application in solution processed OLEDs. The crystal results for three compounds and other detail characteristics will be useful for the readers. Therefore, I recommend this manuscript to Materials after minor revision.

1.     Page 1 line 19-20 : Author did not mention the OLED efficiency such as EQE or current efficiency or power efficiency.

in the noticeable increase of the OLED luminance and efficiency. As a result, the brightness

=> in the noticeable increase of the OLED luminance. As a result, the maximum brightness

2.     Page 2 line 52 : The expressions of “In ref [15],” and “while in ref. [16],” are unfamiliar in the journal paper. “Mao et al. reported that ~~~ [15]” will be better.

3.     Page 4 Figure 1. a)

In the chemical structure, the “( )3” for the general form of diketone was missing.

4.     Pgae 5. Table 1

Luminescent properties of Eu(dik)3DPPZ

=> Photoluminescent properties of the films with Eu(dik)3DPPZ

What is the meaning of “6TCTA:3OXD-7”? If the blend ratio is 1:6:3 for Eu(tta)3DPPZ:TCTA:OXD-7, revise the table 1 with “composition of film” and “ratio”.

5.     Some expression of “Luminescence” should be changed to “Photoluminescence”.

6.     I recommend to put “Figure S8. Electroluminescence spectra of OLEDs S1-S3” between Figure 5 and 6. It is important data. Also if possible, CIE coordinates (x, y) of these spectra would be mentioned.

7.     Page 7. Line 206 : “Current efficiency” is wrong expression.

Time dependence of current efficiency (cd/m2) at 7V (~10 cd/m2) and 8.6V (~100 cd/m2).

=> Lifetime of solution-processed OLED based on Eu(dbm)3DPPZ under the constant voltage of 7V (~10 cd/m2) and 8.6V (~100 cd/m2)

8. Page 7. Line 224 : at 100 cd/A => at 100 cd/m2

Author Response

Dear Reviewer,

we thank you for your attentive reading and helpful comments on this work. Below is a list of changes that we made based on your suggestions.

  1. 1. Page 1 line 19-20 : Author did not mention the OLED efficiency such as EQE or current efficiency or power efficiency.

in the noticeable increase of the OLED luminance and efficiency. As a result, the brightness => in the noticeable increase of the OLED luminance. As a result, the maximum brightness

Corrected.

  1. 2. Page 2 line 52 : The expressions of “In ref [15],” and “while in ref. [16],” are unfamiliar in the journal paper. “Mao et al. reported that ~~~ [15]” will be better.

Corrected

3. Page 4 Figure 1. a)

In the chemical structure, the “( )3” for the general form of diketone was missing.

Corrected.

4. Page 5. Table 1

Luminescent properties of Eu(dik)3DPPZ

=> Photoluminescent properties of the films with Eu(dik)3DPPZ

Corrected.

What is the meaning of “6TCTA:3OXD-7”? If the blend ratio is 1:6:3 for Eu(tta)3DPPZ:TCTA:OXD-7, revise the table 1 with “composition of film” and “ratio”.

Thank you for mentioning! We changed the column title “host” for “host in Eu(dik)3DPPZ:host”, and changed “CBP” for “3CBP”; we also changed the table title, adding “Eu(dik)3DPPZ:host films”. We hope it will be clear now.

5. Some expression of “Luminescence” should be changed to “Photoluminescence”.

Done

6. I recommend to put “Figure S8. Electroluminescence spectra of OLEDs S1-S3” between Figure 5 and 6. It is important data. Also if possible, CIE coordinates (x, y) of these spectra would be mentioned.

We preferred to keep Fig S8 in ESI, since all the Eu spectra are absolutely similar, and this seems a waste of space to redemonstrate them. We have rather added:

All the diodes demonstrated the pure EL of the europium ion (Fig. S8), which coincided exactly with their photoluminescence spectra (Fig. 3);

7. Page 7. Line 206 : “Current efficiency” is wrong expression.

Time dependence of current efficiency (cd/m2) at 7V (~10 cd/m2) and 8.6V (~100 cd/m2).

=> Lifetime of solution-processed OLED based on Eu(dbm)3DPPZ under the constant voltage of 7V (~10 cd/m2) and 8.6V (~100 cd/m2)

Corrected.

8. Page 7. Line 224 : at 100 cd/A => at 100 cd/m2

Corrected.

Reviewer 3 Report

                  This manuscript reports the luminescence characteristics of europium-based mixed-ligand complex. The authors report the application photo-, electro-luminescence and lifetime of electroluminescence. The investigation and conclusion described in this study are relatively poor, so much improvement is needed. I recommend a publication after major revision with additional data and a better structuring of the findings.

1.      The English expression is quite awkward. Correction by native speaker is required. The use of periods and commas should be precise. For example, the decimal representation of the numbers shown in Table 1 should be a period instead of a comma.

2.      The luminescent properties of the europium compound synthesized by the authors were significantly lower than the values reported in other papers. For example, the PLQY reported by other researchers is as high as 91% [Inorg. Chem. 57, 7512 -7515 (2018)], the highest EL efficiency is approximately 3.11% [J. Mater. Chem. C. 5, 12182 - 12188 (2017)], and the optimal red emission CIE coordinates is (0.662, 0.321) [J. Mater. Chem. C 8, 9816–9827 (2020)].

3.      Since the expression of EL data reported by the authors is different from the typical method, it should be changed to an appropriate manner. Basically, The L-J characteristics of devices are essential [i.e. Luminance (cd/m2) vs current density (mA/cm2)] and the external quantum efficiency vs current density plot is also a necessary plot to display device characteristics. .

4.      The authors measured the luminescence characteristics with varying the different ratio of compositions in Table 2, but they did not explain why these values changed, and there was no explanation for the change when the solvent was changed at the same composition ratio. Authors should provide an explanation for these changes.

5.      The authors used only solvents with relatively high volatility, but if there is no major problem with solubility in the solution process, using a solvent such as dichlorobenzene with low volatility can improve the film properties and improve the luminous properties. It is recommended to measure the luminescent properties after forming the film using a low volatility solvent.

6.      Usually, when expressing EL characteristics, the expression (reached up to 700 cd/m2) indicated by the authors is not used, but EL efficiency and luminous CIE coordinates are usually indicated. The authors should provide such information.

Author Response

Dear Reviewer,

we thank you for your attentive reading and helpful comments on this work. Below is a list of changes that we made based on your suggestions.

  1. 1. The English expression is quite awkward. Correction by native speaker is required. The use of periods and commas should be precise. For example, the decimal representation of the numbers shown in Table 1 should be a period instead of a comma.

Thank you, the MS was revised by a native speaker.

  1. 2. The luminescent properties of the europium compound synthesized by the authors were significantly lower than the values reported in other papers. For example, the PLQY reported by other researchers is as high as 91% [Inorg. Chem. 57, 7512 -7515 (2018)], the highest EL efficiency is approximately 3.11% [J. Mater. Chem. C. 5, 12182 - 12188 (2017)], and the optimal red emission CIE coordinates is (0.662, 0.321) [J. Mater. Chem. C 8, 9816–9827 (2020)].

We have also reported Eu complexes with PLQYs over 90% (i.e. http://dx.doi.org/10.1039/C8CC02930J), however, the high PLQY is usually associated with long lifetimes, which are, as we recently demonstrated (http://dx.doi.org/10.1039/d1dt02269e), the key parameter responsible for the low brightness of lanthanide-based OLEDs. Indeed, the brightest Ln-based OLEDs are obtained for the complexes with short lifetimes, even if the PLQYs are also low.

  1. 3. Since the expression of EL data reported by the authors is different from the typical method, it should be changed to an appropriate manner. Basically, The L-J characteristics of devices are essential [i.e. Luminance (cd/m2) vs current density (mA/cm2)] and the external quantum efficiency vs current density plot is also a necessary plot to display device characteristics.

Thank you, we added L-J dependence and CE dependence on V.

  1. 4. The authors measured the luminescence characteristics with varying the different ratio of compositions in Table 2, but they did not explain why these values changed, and there was no explanation for the change when the solvent was changed at the same composition ratio. Authors should provide an explanation for these changes.

Thank you for the question. Regarding the solvent, the explanation was given:

This is likely due to the quenching by the residual solvent, which remains within the film even after the thermal treatment.

We also added the explanation regarding the host change:

Such a dependence may be associated with the participation of the host in the sensitization process. Indeed, as DPPZ demonstrates low sensitization efficiency and even quenching of the Eu luminescence, the presence of the host, able to sensitize its luminescence, may affect the luminescence efficiency.

  1. 5. The authors used only solvents with relatively high volatility, but if there is no major problem with solubility in the solution process, using a solvent such as dichlorobenzene with low volatility can improve the film properties and improve the luminous properties. It is recommended to measure the luminescent properties after forming the film using a low volatility solvent.

Since the presence of the residual solvent, remaining within the film even after the thermal treatment, is usually considered as the major problem of the solution-based OLEDs, the highly-volatile solvents are preferable to increase the solution-processed OLED performance.

  1. 6. Usually, when expressing EL characteristics, the expression (reached up to 700 cd/m2) indicated by the authors is not used, but EL efficiency and luminous CIE coordinates are usually indicated. The authors should provide such information.

As we mentioned in our paper, the electroluminescence spectra of all obtained OLEDs are the same, remain constant over time, and correspond to pure europium luminescence, which is not affected by host ratio and solvent selection. That is why CIE coordinates correspond to europium emission (x=0.66, y=0.32), and actually, they are usually not mentioned for Ln-based OLEDs.

We added the statement:

All the diodes demonstrated the pure EL of the europium ion (Fig. S8) with CIE coordinates (x=0.66, y=0.32),

Round 2

Reviewer 3 Report

The external quantum efficiency vs current density plot is still missing. Since quantum efficiency is very important, the authors should include it in their manuscript.

Author Response

Dear Reviewer, unfortunately, our electroluminescence measurement installation still does not include the integration sphere which is needed for the proper measurements of the EQE. If the Reviewer insists, we can indirectly evaluate these values, but we would kindly ask to keep only CE-V dependence, which was properly measured and also contains information on the OLED efficiency. Thus, presently, we resubmitted the MS in its previous form.

We are doing our best to improve our equipment to be able to measure EQE in future.